# TILDE-Q: A Transformation Invariant Loss Function for Time-Series Forecasting

## Abstract

Time-series forecasting has gained increasing attention in the field of artificial intelligence due to its potential to address real-world problems across various domains, including energy, weather, traffic, and economy. While time-series forecasting is a well-researched field, predicting complex temporal patterns such as sudden changes in sequential data still poses a challenge with current models. This difficulty stems from minimizing $L_p$ norm distances as loss functions, such as mean absolute error (MAE) or mean square error (MSE), which are susceptible to both intricate temporal dynamics modeling and signal shape capturing. Furthermore, these functions often cause models to behave aberrantly and generate uncorrelated results with the original time-series. Consequently, the development of a shape-aware loss function that goes beyond mere point-wise comparison is essential. In this paper, we examine the definition of shape and distortions, which are crucial for shape-awareness in time-series forecasting, and provide a design rationale for the shape-aware loss function. Based on our design rationale, we propose a novel, compact loss function called TILDE-Q (Transformation Invariant Loss function with Distance EQuilibrium) that considers not only amplitude and phase distortions but also allows models to capture the shape of time-series sequences. Furthermore, TILDE-Q supports the simultaneous modeling of periodic and nonperiodic temporal dynamics. We evaluate the efficacy of TILDE-Q by conducting extensive experiments under both periodic and nonperiodic conditions with various models ranging from naive to state-of-the-art. The experimental results show that the models trained with TILDE-Q surpass those trained with other metrics, such as MSE and DILATE, in various real-world applications, including electricity, traffic, economics, weather, and electricity transformer temperature (ETT).

## 1 Introduction

Time-series forecasting has been a core problem across various domains, including traffic domain (Li et al., 2018; Lee et al., 2020), economy (Zhu & Shasha, 2002), and disease propagation analysis (Matsubara et al., 2014). One of the key challenges in time-series forecasting is the modeling of complex temporal dynamics (e.g., non-stationary signal and periodicity). Temporal dynamics, intuitively, shape, is the most emphasized keywords in time-series domains, such as rush hour of traffic data or abnormal usage of electricity (Keogh et al., 2003; Bakshi & Stephanopoulos, 1994; Weigend & Gershenfeld, 1994; Wu et al., 2021; Zhou et al., 2022).

Although deep learning methods are an appealing solution to model complex non-linear temporal dependencies and nonstationary signals, recent studies have revealed that even deep learning is often inadequate to model temporal dynamics. To properly model temporal dynamics, novel deep learning approaches, such as Autoformer (Wu et al., 2021) and FEDFormer (Zhou et al., 2022), have proposed input sequence decomposition. Still, they are trained with $L_p$ norm-based loss function, which could not properly model the temporal dynamics, as shown in Fig. 1, (top). On the other hand, Le Guen & Thome (2019) attempt to model sudden changes in a timely and accurate manner with dynamic time warping (DTW), and Bica et al. (2020) adopt domain adversarial training to learn balanced representations, which is a treatment invariant representations over time. Le Guen & Thome (2019); Bica et al. (2020) try to capture the shape but still have some limitations, as depicted in Fig. 1 (middle), implying the need for further investigation of the shape.

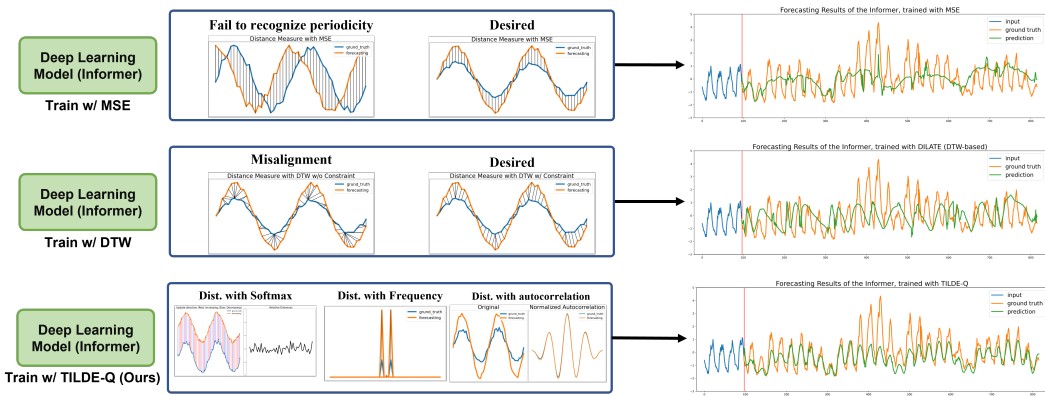

Figure 1: Ground-truth and forecasting results of Informer model with three training metrics, as shown in the blue box: (top) MSE, (middle) DTW-based, and (bottom) TILDE-Q loss function. (top, middle) The blue boxes indicates the original intention of loss function (desired) and misbehaviors.

The identification of **shape**, denoting the pattern in time-series data within a given time interval, plays an important role in addressing aforementioned limitation in time-series forecasting problem. It can provide valuable information, such as rise, drop, trough, peak, and plateau. We refer to the prediction as *informative* when it can appropriately model the shape. In real-world applications, including economics, informative prediction is invaluable for decision-making. To achieve such informative forecasting, a model should account for shape instead of solely aiming to forecast accurate value for each time step. However, existing methods inadequately consider the shape (Wu et al., 2021; Zhou et al., 2022; Bica et al., 2020; Le Guen & Thome, 2019). Moreover, deep learning model tends to opt for an *easy learning path* (Karras et al., 2019), yielding inaccurate and uninformative forecasting results disregarding the characteristics of time-series data. Fig. 1 illustrates three real forecasting results obtained with Informer (Zhou et al., 2021) and different training metrics. When the mean squared error (MSE) is used as an objective, the model aims to reduce the gap between prediction and ground truth for each time-step. This "point-wise" distance-based optimization has less ability to model shape, resulting in generating uninformative predictions regardless of temporal dynamics (Fig. 1 (top)); the model rarely provides information about the time-series. In contrast, if both gap and shape of the prediction and ground truth are taken into account, the model can achieve high accuracy with proper temporal dynamics, as shown in Fig. 1 (bottom). Consequently, time-series forecasting requires a loss function that consider both point-wise distance (i.e., traditional goal) and shape.

In this work, we aim to design a novel objective function that guides models in improving forecasting performance by learning shapes in time-series data. To design a shape-aware loss function, we review existing literature (Esling & Agon, 2012; Bakshi & Stephanopoulos, 1994; Keogh, 2003) and explore the concepts of *shapes* and *distortions* that impede appropriate measurement of similarity between two time-series data in terms of shapes (Sec. 3.1, Sec. 3.2, and Sec. 3.3). Based on our investigation, we propose the necessary conditions for constructing an objective function for shape-aware time-series forecasting (Sec. 4.1). Subsequently, we present a novel loss function, TILDE-Q (Transformation Invariant Loss function with Distance EQualibrium), which enables shape-aware representation learning by utilizing three loss terms that are invariant to distortions (Sec. 4.2). For evaluation, we conduct extensive experiments with state-of-the-art deep learning models with TILDE-Q. The experimental results indicate that TILDE-Q is model-agnostic and outperforms MSE and DILATE in MSE and shape-related metrics.

**Contributions** In summary, our study makes the following contributions. (1) We delve into the concept of shape awareness and distortion invariances in the context of time-series forecasting. By thoroughly investigating these distortions, we enhance our understanding of their impact on time-series forecasting problems. (2) We propose and implement TILDE-Q, which has invariances to three distortions and achieves shape-awareness, empowering informative forecasting in a timely manner. (3) We empirically demonstrate that the proposed TILDE-Q allows models to have higher accuracy compared to the models trained with other existing metrics, such as MSE and DILATE.

## 2 RELATED WORK

### 2.1 TIME-SERIES FORECASTING

Many time-series forecasting methods are available, ranging from traditional models, such as ARIMA model (Box et al., 2015) and hidden Markov model (Pesaran et al., 2004), to recent deep learning models. In this section, we briefly describe the recent deep learning models for time-series forecasting. Motivated by the huge success of recurrent neural networks (RNNs) (Clevert et al., 2016; Li et al., 2018; Yu et al., 2017), many novel deep learning architectures have been developed for improving forecasting performance. To effectively capture long-term dependency, which is a limitation of RNNs, Stoller et al. (2020) have proposed convolutional neural networks (CNNs). However, it is required to stack lots of the same CNNs to capture long-term dependency (Zhou et al., 2021). Attention-based models, including Transformer (Vaswani et al., 2017) and Informer (Zhou et al., 2021), have been another popular research direction in time-series forecasting. Although these models effectively capture temporal dependencies, they incur high computational costs and often struggle to obtain appropriate temporal information (Wu et al., 2021). To cope with the problem, Wu et al. (2021); Zhou et al. (2022) have adopted the input decomposition method, which helps models better encode appropriate information. Other state-of-the-art models adopt neural memory networks (Kaiser et al., 2017; Sukhbaatar et al., 2015; Madotto et al., 2018; Lee et al., 2022), which refer to historical data stored in the memory to generate meaningful representation.

### 2.2 TRAINING METRICS

Conventionally, mean squared error (MSE), $L_p$ norm and its variants are mainstream metrics used to optimize forecasting models. However, they are not optimal for training forecasting models (Esling & Agon, 2012) because the time-series is temporally continuous. Moreover, the $L_p$ norm provides less information about temporal correlation among time-series data. To better model temporal dynamics in time-series data, researchers have used differentiable, approximated dynamic time warping (DTW) as an alternative metric of MSE (Cuturi & Blondel, 2017; Abid & Zou, 2018; Mensch & Blondel, 2018). However, using DTW as a loss function results in temporal localization of changes being ignored. Recently, Le Guen & Thome (2019) have suggested DILATE, a training metric to catch sudden changes of nonstationary signals in a timely manner with smooth approximation of DTW and penalized temporal distortion index (TDI). To guarantee DILATE's operation in a timely manner, penalized TDI issues a harsh penalty when predictions showed high temporal distortion. However, the TDI relies on the DTW path, and DTW often showed misalignment because of noise and scale sensitivity. Thus, DILATE often loses its advantage with complex data, showing disadvantages at the training. In this paper, we discuss distortions and transformation invariances and design a new loss function that enables models to learn shapes in the data and produce noise-robust forecasting results.

## 3 PRELIMINARY

In this section, we investigate common distortions focusing on the goal of time-series forecasting (i.e., modeling temporal dynamics and accurate forecasting). To clarify the concepts of time-series forecasting and related terms, we first define the notations and terms used (Sec. 3.1). We then discuss common distortions in time-series from the transformation perspective that need to be considered for building a shape-aware loss function (Sec. 3.2) and describe how other loss functions (e.g., dynamic time warping (DTW) and temporal distortion index (TDI)) handle shapes during learning (Sec. 3.3). We will discuss the conditions for effective time-series forecasting in the next session (Sec. 4.1).

### 3.1 NOTATIONS AND DEFINITIONS

Let $X_t$ denote a data point at a time step $t$. We define a time-series forecasting problem as follows:

**Definition 3.1.** *Given $T$-length historical time-series $\mathbf{X} = [X_{t-T+1}, \ldots, X_t], X_i \in \mathbb{R}^F$ at time $i$ and a corresponding $T'$-length future time-series $\mathbf{Y} = [Y_{t+1}, \ldots, Y_{t+T'}], Y_i \in \mathbb{R}^C$, time-series forecasting aims to learn the mapping function $f : \mathbb{R}^{T \times F} \to \mathbb{R}^{T' \times C}$.*

To distinguish between the label (i.e., ground truth) and prediction time-series data, we note the label data as $\mathbf{Y}$ and prediction data as $\hat{\mathbf{Y}}$. Next, we set up two goals for time-series forecasting, which

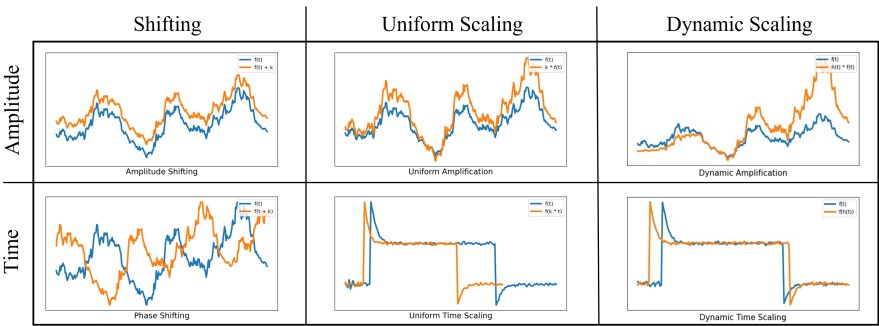

Figure 2: Example of the six distortions on the amplitude axis (top) and temporal axis (bottom).

require not only precise but also informative forecasting (Wu et al., 2021; Zhou et al., 2022; Le Guen & Thome, 2019) as follows:

- The mapping function $f$ should be learnt to point-wisely reduce distance between $\hat{\mathbf{Y}}$ and $\mathbf{Y}$;
- The output $\hat{\mathbf{Y}}$ should have similar temporal dynamics with $\mathbf{Y}$.

Temporal dynamics are informative patterns in a time-series, such as rise, drop, peak, and plateau. The optimization for point-wise distance reduction is a conventional method used in the deep learning domain, which can be obtained using the MAE or MSE. However, in a real-world problem, such as traffic speed or stock market prediction, accurate forecasting of temporal dynamics is required. Esling & Agon (2012) also emphasized the measurement of temporal dynamics, as *"...allowing the recognition of perceptually similar objects even though they are not mathematically identical."* In this paper, we define temporal dynamics as follows:

**Definition 3.2.** *Temporal dynamics (or shapes) are informative periodic and nonperiodic patterns in time-series data.*

In this work, we aim to design a shape-aware loss function that satisfies both goals. To this end, we first discuss distortions that two time-series with similar shapes can have.

**Definition 3.3.** *Given two time-series $\mathbf{F}$ and $\mathbf{G}$ having similar shapes but not being mathematically identical, let $\mathcal{H}$ is transformation that satisfies $\mathbf{F} = \mathcal{H}(\mathbf{G})$. Then, the time-series $\mathbf{F}$ and $\mathbf{G}$ are considered to have a distortion, which can be represented by the transformation $\mathcal{H}$.*

A distortion can generally be classified as a temporal distortion (i.e., *warping*) or an amplitude distortion (i.e., *scaling*) depending on its dimension–time and amplitude. Existing distortions in the data lead to misbehavior of the model, as they distort the measurements to be inaccurate. For example, if we have two time-series $\mathbf{F}$ and $\mathbf{G} = \mathbf{F} + k$, which have similar shapes but different means, $\mathbf{G}$ could represent many temporal dynamics of $\mathbf{F}$. However, measurements often evaluate $\mathbf{F}$ and $\mathbf{G}$ as completely different signals and cause misguidance of the model in training (e.g., measuring the distance of $\mathbf{F}$ and $\mathbf{G}$ with MSE). As such, it is important to have measurements that consider a similar shape invariant to distortion. We define a measurement for distortion as:

**Definition 3.4.** *Let transformation $\mathcal{H}$ represent a distortion $H$. Then, we call measurement $\mathcal{D}$ invariant to $\mathcal{H}$ if $\exists \delta > 0 : \mathcal{D}(\mathbf{T}, \mathcal{H}(\mathbf{T})) < \delta$ for any time-series $\mathbf{T}$.*

## 3.2 Time-Series Distortions in Transformation Perspectives

Distortion, a gap between two similar time-series, affects shape capturing in time-series data. Thus, it is important to investigate different distortions and their impacts on representation learning aspects. There are six common time-series distortions that models encounter during learning (Esling & Agon, 2012; Batista et al., 2014; Berkhin, 2006; Warren Liao, 2005; Kerr et al., 2008)–Amplitude Shifting, Phase Shifting, Uniform Amplification, Uniform Time Scaling, Dynamic Amplification, and Dynamic Time Scaling. Next, we explain each common time-series distortion in terms of transformation with an $n$-length time-series $\mathbf{F}(t) = [f(t_1), f(t_2), \dots, f(t_n)]$, where t $= [t_1, t_2, \dots, t_n]$. Fig. 2 presents example distortions, categorized by amplitude and time dimensions.

- *Amplitude Shifting* describes how much a time-series shifts against another time-series. This can be described with two time-series and the degree of shifting $k$: $\mathbf{G}(t) = \mathbf{F}(t) + k = [f(t_1) + k, \ldots, f(t_n) + k]$, where $k \in \mathbb{R}$ is constant.

- *Phase Shifting* is the same type of transformation (i.e., translation) as amplitude shifting, but it occurs along the temporal dimension. This distortion can be represented by two time-series functions with the degree of shift $k$: $\mathbf{G}(t) = \mathbf{F}(t + k) = [f(t_1 + k), \ldots, f(t_n + k)]$, where $k \in \mathbb{R}$ is constant. Cross-correlation (Paparrizos & Gravano, 2015; Vlachos et al., 2005) is the most popular measure method that is invariant to this distortion.

- *Uniform Amplification* is a transformation that changes the amplitude by multiplication of $k \in \mathbb{R}$. This distortion can be described with two functions and a multiplication factor $k$: $\mathbf{G}(t) = k \cdot \mathbf{F}(t) = [k \cdot f(t_1), \ldots, k \cdot f(t_n)]$.

- *Uniform Time Scaling* refers to a uniformly shortened or lengthened $\mathbf{F}(t)$ on the temporal axis. This distortion can be represented as $\mathbf{G}(t) = [g(t_1), \ldots, g(t_m)]$, where $g(t_i) = f(t_{\lceil k \cdot i \rceil})$ and $k \in \mathbb{R}^+$. Although Keogh et al. (2004) have proposed uniform time warping methods to handle this distortion, it still remains a challenging distortion type to measure because of the difficulty in identifying the scaling factor $k$ without testing all possible cases (Keogh, 2003).

- *Dynamic Amplification* is any distortion that occurs through non-zero multiplication along the amplitude dimension. This distortion can be described as follows: $\mathbf{G}(t) = \mathbf{H}(t) \cdot \mathbf{F}(t) = [h(t_1) \cdot f(t_1), \ldots, h(t_n) \cdot f(t_n)]$ with function $h(t)$, such that $\forall_{t \in \mathbb{T}}, h(t) \neq 0$. Local amplification is representative of such distortions, which still remains challenging to solve.

- *Dynamic Time Scaling* refers to any transformation that dynamically lengthens or shortens signals along the temporal dimension, including local time scaling (Batista et al., 2014) and occlusion (Batista et al., 2014; Vlachos et al., 2003). It can be represented as follows: $\mathbf{G}(t) = \mathbf{F}(h(t)) = [f(h(t_1)), \ldots, f(h(t_n))]$, where $h(t)$ is a positive, strictly increasing function. DTW (Bellman & Kalaba, 1959; Berndt & Clifford, 1994; Keogh & Ratanamahatana, 2005) is the most popular technique invariant to this distortion. Das et al. (1997) have also introduced the longest common subsequence (LCSS) algorithm to tackle occlusion, noise, and outliers in this distortion.

Shape-aware clustering (Bellman & Kalaba, 1959; Batista et al., 2014; Paparrizos & Gravano, 2015; Berkhin, 2006; Warren Liao, 2005; Kerr et al., 2008) and classification (Xi et al., 2006; Batista et al., 2014; Srisai & Ratanamahatana, 2009) tasks that consider shapes have been extensively studied. However, only a few studies exist for time-series forecasting tasks, including Le Guen & Thome (2019) that utilize DTW and TDI for modeling temporal dynamics. Next, we describe the MSE and DILATE, proposed by Le Guen & Thome (2019), and discuss their invariance to distortions.

### 3.3 Distortion Handling in Current Time-Series Forecasting Objectives

Many measurement metrics have been used in the time-series forecasting domain, and those based on the $L_p$ distance, including Euclidean distance, are widely used to handle time-series data. However, such metrics are not invariant to the aforementioned distortions (Ding et al., 2008; Le Guen & Thome, 2019) because of their point-wise mapping. In particular, since $L_p$ distance compares the values per time step, it cannot handle temporal distortions appropriately and is vulnerable to data scaling. Le Guen & Thome (2019) have proposed a loss function called DILATE to overcome the inadequate characteristic in the $L_p$ distance metric by recognizing temporal dynamics with DTW and TDI. In terms of transformation, DILATE handles dynamic time scaling, especially local time scaling with DTW, and phase shifting with penalized TDI, defined as follows:

$$\mathcal{L}_{DILATE}(\hat{y}_i, y_i) := -\gamma \log \Big( \sum_{\mathbf{A} \in \mathcal{A}_{k,k}} e^{-\frac{\langle \mathbf{A}, \alpha \Delta(\hat{y}_i, y_i) + (1-\alpha)\mathbf{\Omega} \rangle}{\gamma}} \Big),$$

where $\mathbf{A}$, $\Delta(\hat{y}_i, y_i)$, $\mathbf{\Omega}$ are the warping path, cost matrix, and penalization matrix, respectively.

While DILATE performs better than existing methods, it has a limitation from the perspective of invariance. DILATE highly depends on DTW, which allows for the dynamic alignment of the time-series for a predefined window. In such windows, DTW can align the signal regardless of its information (e.g., periodicity). As a result, the model creates misbehavior that can cheat DTW within

the window, as shown in Fig. 1 middle. DTW's scale and noise sensitivity are also problematic. DTW computes the Euclidean distance of two time-series after its temporal alignment in dynamic programming, and the alignment relies on the distance function. Consequently, the dynamic alignment of DTW can be properly achieved only when the two time-series have the same range (Esling & Agon, 2012; Bellman & Kalaba, 1959). This means that it hardly achieves invariance to amplitude distortion without appropriate pre-processing. Gong & Chen (2017) also show that DTW poorly matches the prediction and target (i.e., ground truth) time-series with amplitude shifting. Even when the target time-series is aligned with normalization, the appropriate alignment of the prediction and target time-series cannot be guaranteed because of DTW's high sensitivity to noise. As a result, DILATE can generate poor alignment results, which can cause wrong TDI optimization, producing incorrect results and instability during the optimization steps. To design an effective shape-aware loss function, we must understand the measures and in which cases they have transformation invariances. In the next section, we interpret transformations from a time-series forecasting viewpoint and discuss the types of transformations that should be considered in objective function design.

## 4 METHODS

In this section, we discuss and propose the design rationale for the shape-aware loss function (Sec. 4.1). Based on the design rationale, we implement a novel loss function, TILDE-Q (a Transformation Invariant Loss function with Distance EQuilibrium), which allows models to perform shape-aware time-series forecasting based on three distortion invariances.

### 4.1 TRANSFORMATION INVARIANCES IN TIME-SERIES FORECASTING

In the time-series domain, data often have various distortions; thus, measurements need to satisfy numerous transformation invariances for meaningfully modeling temporal dynamics. As discussed in Sec. 3.1, we set the goals of time-series forecasting as (1) point-wisely reducing the gap between the prediction and target time-series and (2) preserving the temporal dynamics of the target time-series. To satisfy both of them, we have to consider (1) a method that does not negatively impact on the traditional goal of accurate time-series forecasting and (2) distortions that play a crucial role in capturing the temporal dynamics of the target time-series. In this section, we review all six distortions based on whether their corresponding invariance is feasible to be a loss function for time-series forecasting, discuss the loss function's benefits and trade-offs, and identify appropriate distortions to be considered in time-series forecasting.

**Amplitude Shifting** In a wide range of situations, it is beneficial to capture the trends of time-series sequences despite shifts in amplitude. Thus, being invariant to amplitude shifting in a loss function is highly advantageous in time-series forecasting: (1) shape awareness invariant to amplitude shifting, (2) accurate deviation of values in modeling, and (3) effective on-time prediction of the peak or sudden changes. To guarantee an amplitude shifting invariance in the optimization stage, the loss function should induce an equal gap $k$ between the prediction and ground truth data in each step. Specifically, the loss function considering amplitude shifting should satisfy:

$$\mathcal{L}(\mathbf{Y}, \hat{\mathbf{Y}}) = 0 \Leftrightarrow \forall_{i \in [1,\dots,n]}, d(y_i, \hat{y}_i) = k, \tag{1}$$

where $k \in \mathbb{R}$ is an arbitrary and equal gap, and $d(y_i, \hat{y}_i)$ is a signed distance with a boundary $y_i > \hat{y}_i$. By allowing tolerance between the prediction and target time-series, models can follow trends in time-series instead of predicting exact values point-wisely. In short, unlike existing loss functions, which handle only point-wise distance (e.g., DTW), we should deal with both point-wise distance and its relational distance values to guarantee amplitude shifting.

**Phase Shifting** There are some forecasting tasks whose main objectives concern accurate forecasting of peaks and periodicity in time-series (e.g., heartbeat data and stock price data). For such tasks, phase shifting invariance is an optimal solution for (1) modeling periodicity, regardless of the translation on the temporal axis, and (2) having precise statistics with shapes, such as peak and plateau values. To be invariant to phase shifting, the loss function should satisfy

$$\mathcal{L}(\mathbf{Y}, \hat{\mathbf{Y}}) = 0 \Leftrightarrow \mathbf{Y} \text{ and } \hat{\mathbf{Y}} \text{ have the same dominant frequency.} \tag{2}$$

Note that Eq. 2 allows a similar shape as the target time-series in forecasting, not exactly the same shape (e.g., $\sin(x)$ and $2\sin(x + x_0)$ with the same dominant frequency).

**Uniform Amplification** This proposition can be utilized in the case of sparse data that contains a significant number of zeros. By adopting uniform amplification invariance, models are able to focus on non-zero sequences, whereas this proposition allows models to receive less penalty in zero sequences. Since it guarantees shape awareness with a multiplication factor in a timely manner, as shown in Fig. 2, invariance for uniform amplification fits well. To have a model trained with uniform amplification invariance, the loss function should satisfy the following proposition:

$$\mathcal{L}(\mathbf{Y}, \hat{\mathbf{Y}}) = 0 \Leftrightarrow \forall_{i \in [1,\ldots,n]}, \frac{y_i}{\hat{y}_i} = k(\hat{y}_i \neq 0). \tag{3}$$

**Uniform Time Scaling, Dynamic Amplification, and Dynamic Time Scaling** After careful consideration, we conclude that uniform time scaling, dynamic amplification, and dynamic time scaling are incompatible for optimization. the reasons are described below.

To achieve invariance for uniform time scaling, the loss function should satisfy below:

$$\mathcal{L}(\mathbf{Y}, \hat{\mathbf{Y}}) = 0 \Leftrightarrow \exists c \in \mathbb{Z}^+ : \{c | y_i = \hat{y}_{ci}\} \cup \{c | y_{ci} = \hat{y}_i\} \forall i \in [0, 1, \ldots, T'].$$

This proposition will negatively influence the original temporal dynamics, considering that it creates the tolerance for mispredicting periodicity (e.g., daily periodic signals) and cannot identify events (e.g., abrupt changing values) in a timely manner. In summary, it hinders models from capturing shapes and corrupts periodic information.

For both dynamic amplification and time scaling, the loss functions are zero for all pairs when there is no limit for tolerance. Formally, the proposition for dynamic amplification invariance is as follows:

$$\mathcal{L}(\mathbf{Y}, \hat{\mathbf{Y}}) = 0 \Leftrightarrow \forall c_i \in \mathbb{R} : y_i = c_i \hat{y}_i,$$

If a loss function satisfies this proposition without bound for $c_i$, it is always zero because there always exists $c_i = y_i / \hat{y}_i$, except $\hat{y}_i = 0$. Therefore, it is not able to provide any information because all random values could be an optimal solution. The same situation happens for the dynamic time scaling if we do not limit the window. Consequently, all three objectives–uniform time scaling, dynamic amplification, and dynamic time scaling are unsuitable to be objectives in time-series forecasting.

## 4.2 TILDE-Q: Transformation Invariant Loss function with Distance EQuilibrium

To build a transformation invariant loss function, we need to design a loss function that satisfies the proposition for amplitude shifting (Eq. 1), phase shifting (Eq. 2), and uniform amplification shifting invariance (Eq. 3), as discussed in Sec. 4.1. Furthermore, the loss function should guarantee a small $L_p$ norm between prediction and label, which is the traditional goal of forecasting. Both conditions are hard to simultaneously satisfy by existing loss functions, such as the MSE or DILATE. To handle all three distortions while considering traditional goal, we build three objective functions (*a.shift*, *phase*, and *amp* losses) that can achieve one or more invariance by using softmax, Fourier coefficient, and autocorrelation to design a loss function.

**Amplitude Shifting Invariance with Softmax (Amplitude Shifting)** To strengthen amplitude shifting invariance, we design a loss function that satisfies Eq. 1. This means that $d(y_i, \hat{y}_i)$ must have the same value for all $i$. To satisfy this condition, we utilize the softmax function:

$$\mathcal{L}_{a.shift}(\mathbf{Y}, \hat{\mathbf{Y}}) = T' \sum_{i=1}^{T'} |\frac{1}{T'} - \text{Softmax}(d(y_i, \hat{y}_i))|, \text{Softmax}(d(y_i, \hat{y}_i)) = \frac{e^{d(y_i, \hat{y}_i)}}{\sum_{j=1}^{T'} e^{d(y_j, \hat{y}_j)}} \tag{4}$$

where $T'$, Softmax, and $d(\cdot, \cdot)$ are the sequence length, softmax function, and signed distance function, respectively. Because softmax produces the proportion of each value, it can obtain the optimal solution only when it satisfies Eq. 1. Since Softmax outputs the relative values, it could handle any gap $k$.

**Invariances with Fourier Coefficients (Phase Shifting)** As discussed in Sec. 4.1, a potential method that can be used to obtain phase shifting invariance is the use of Fourier coefficients. According to the literature (NG & GOLDBERGER, 2007), the original time-series can be reconstructed with a few dominant frequencies. Thus, we utilize the gap between dominant Fourier coefficients of

Table 1: Experimental results on six real-world datasets (four cases) with four the state-of-the-art models and three training metrics. For all experiment, we set input sequence length $T = 96$.

| Model | | N-Beats | | | | | | Informer | | | | | | Autoformer | | | | | | FEDformer | | | | | |
|---|---|---|---|---|---|---|---|---|---|---|---|---|---|---|---|---|---|---|---|---|---|---|---|---|---|---|
| Methods | | MSE | | DILATE | | TILDE-Q | | MSE | | DILATE | | TILDE-Q | | MSE | | DILATE | | TILDE-Q | | MSE | | DILATE | | TILDE-Q | |
| Metric | | MSE | LCSS | MSE | LCSS | MSE | LCSS | MSE | LCSS | MSE | LCSS | MSE | LCSS | MSE | LCSS | MSE | LCSS | MSE | LCSS | MSE | LCSS | MSE | LCSS | MSE | LCSS |
| ETTh2 | 96 | 0.187 | 0.468 | 0.310 | 0.487 | **0.155** | **0.586** | 0.246 | 0.463 | 0.328 | 0.503 | **0.176** | **0.537** | 0.153 | 0.618 | 0.221 | 0.531 | **0.149** | **0.631** | **0.130** | **0.669** | 0.191 | 0.526 | 0.138 | 0.662 |
| | 192 | 0.239 | 0.450 | 0.618 | 0.463 | **0.173** | **0.581** | 0.281 | 0.425 | 0.408 | **0.489** | 0.243 | 0.431 | **0.197** | **0.601** | 0.282 | 0.533 | 0.207 | 0.598 | **0.182** | **0.623** | 0.269 | 0.526 | 0.199 | 0.612 |
| | 336 | 0.289 | 0.454 | 1.140 | 0.458 | **0.213** | **0.537** | 0.308 | 0.443 | 0.416 | **0.506** | 0.295 | 0.416 | 0.239 | 0.595 | 0.375 | 0.525 | **0.236** | **0.597** | 0.230 | **0.605** | 0.351 | 0.509 | 0.238 | 0.604 |
| | 720 | 0.388 | 0.438 | 1.671 | 0.457 | **0.304** | **0.528** | **0.287** | 0.442 | 0.422 | 0.481 | 0.315 | 0.426 | 0.285 | 0.577 | 0.429 | 0.492 | **0.237** | **0.579** | 0.278 | **0.591** | 0.433 | 0.509 | 0.287 | 0.581 |
| ETTm2 | 96 | **0.079** | 0.672 | 0.152 | 0.437 | 0.095 | **0.690** | 0.088 | 0.738 | 0.126 | 0.512 | **0.087** | **0.781** | 0.099 | 0.675 | 0.113 | 0.593 | **0.094** | **0.707** | 0.068 | 0.787 | 0.115 | 0.632 | **0.067** | **0.792** |
| | 192 | **0.122** | 0.576 | 0.205 | 0.510 | 0.128 | **0.616** | **0.115** | 0.670 | 0.234 | 0.526 | 0.131 | **0.698** | 0.134 | 0.651 | 0.185 | 0.550 | **0.125** | **0.681** | 0.098 | 0.734 | 0.185 | 0.539 | **0.097** | **0.738** |
| | 336 | 0.182 | 0.458 | 0.250 | 0.481 | **0.170** | **0.619** | 0.186 | 0.636 | 0.280 | 0.502 | **0.176** | **0.655** | 0.158 | 0.603 | 0.200 | 0.537 | **0.154** | **0.616** | 0.133 | 0.667 | 0.249 | 0.505 | **0.127** | **0.682** |
| | 720 | 0.237 | 0.492 | 0.417 | 0.583 | **0.233** | **0.707** | 0.216 | 0.576 | 0.374 | 0.474 | **0.206** | **0.586** | 0.199 | 0.606 | 0.266 | 0.500 | **0.188** | **0.627** | 0.196 | 0.626 | 0.291 | 0.481 | **0.182** | **0.636** |
| ECL | 96 | 0.366 | 0.658 | 1.115 | 0.507 | **0.318** | **0.722** | **0.270** | 0.703 | 0.985 | 0.632 | 0.280 | **0.727** | 0.420 | 0.648 | 0.681 | 0.625 | **0.351** | **0.691** | **0.253** | **0.732** | 0.479 | 0.694 | 0.264 | 0.727 |
| | 192 | 0.430 | 0.621 | 1.185 | 0.497 | **0.338** | **0.718** | **0.279** | 0.706 | 1.120 | 0.605 | 0.307 | **0.733** | 0.420 | 0.657 | 0.731 | 0.611 | **0.403** | **0.668** | 0.295 | 0.731 | 0.549 | 0.681 | **0.282** | **0.734** |
| | 336 | 0.519 | 0.596 | 1.246 | 0.509 | **0.383** | **0.711** | **0.320** | **0.722** | 1.233 | 0.569 | 0.327 | 0.714 | **0.462** | **0.653** | 0.789 | 0.609 | 0.463 | 0.642 | **0.331** | 0.721 | 0.697 | 0.689 | 0.339 | **0.730** |
| | 720 | 0.624 | 0.571 | 1.306 | 0.533 | **0.454** | **0.696** | 0.641 | 0.456 | 1.370 | 0.550 | **0.467** | **0.629** | 0.500 | 0.618 | 0.863 | 0.607 | 0.504 | **0.642** | 0.396 | 0.696 | 0.774 | 0.640 | **0.394** | **0.701** |
| Exchange | 96 | 0.450 | 0.442 | 0.394 | 0.432 | **0.275** | **0.447** | 0.353 | **0.469** | 0.326 | 0.468 | 0.526 | 0.455 | 0.247 | 0.458 | 0.192 | **0.465** | **0.173** | 0.458 | 0.144 | 0.435 | 0.388 | 0.444 | **0.122** | **0.470** |
| | 192 | **1.216** | 0.416 | 1.568 | 0.406 | 1.662 | **0.435** | **0.968** | 0.465 | 0.974 | 0.458 | 1.285 | **0.496** | 0.325 | 0.432 | 0.473 | 0.412 | **0.295** | **0.443** | **0.269** | 0.420 | 0.591 | 0.419 | 0.296 | **0.447** |
| | 336 | **1.453** | 0.413 | 3.678 | 0.387 | 1.843 | **0.460** | **1.371** | 0.468 | 1.673 | 0.443 | 1.691 | **0.493** | 0.548 | 0.328 | 0.803 | 0.311 | **0.533** | **0.332** | **0.492** | 0.414 | 0.752 | 0.397 | 0.590 | **0.434** |
| | 720 | **1.856** | 0.407 | 3.901 | 0.340 | 2.849 | **0.462** | **1.764** | 0.468 | 1.829 | **0.529** | 1.913 | 0.510 | 1.362 | 0.236 | 1.494 | 0.230 | **1.199** | 0.223 | 1.212 | 0.384 | 1.511 | 0.376 | **1.170** | **0.393** |
| Traffic | 96 | 0.234 | 0.830 | 2.332 | 0.525 | **0.229** | **0.837** | 0.261 | 0.833 | 2.961 | 0.731 | **0.228** | **0.849** | 0.256 | 0.876 | 0.483 | 0.852 | **0.227** | **0.888** | 0.207 | 0.882 | 0.353 | 0.861 | **0.187** | **0.898** |
| | 192 | **0.301** | 0.792 | 2.563 | 0.552 | 0.335 | **0.803** | 0.292 | 0.816 | 2.998 | 0.739 | **0.275** | **0.825** | 0.260 | 0.878 | 0.565 | 0.819 | **0.250** | **0.882** | 0.205 | 0.895 | 1.468 | 0.859 | **0.196** | **0.898** |
| | 336 | **0.345** | 0.792 | 2.460 | 0.521 | 0.399 | **0.821** | 0.311 | 0.811 | 2.970 | 0.712 | **0.299** | **0.817** | 0.247 | 0.880 | 0.816 | 0.805 | **0.242** | 0.876 | 0.214 | **0.902** | 2.974 | 0.852 | **0.206** | 0.873 |
| | 720 | **0.430** | 0.796 | 2.352 | 0.518 | 0.448 | **0.809** | 0.347 | **0.815** | 2.685 | 0.587 | 0.386 | 0.775 | **0.272** | **0.871** | 1.073 | 0.818 | 0.284 | 0.868 | 0.229 | **0.892** | 3.083 | 0.858 | 0.231 | 0.873 |
| Weather | 96 | 0.004 | 0.407 | 0.002 | 0.426 | **0.001** | **0.517** | 0.004 | 0.456 | 0.007 | 0.516 | **0.002** | **0.560** | 0.017 | 0.482 | 0.002 | 0.531 | **0.001** | **0.546** | 0.007 | 0.526 | 0.002 | 0.554 | **0.001** | **0.579** |
| | 192 | 0.006 | 0.421 | 0.003 | 0.431 | **0.002** | **0.508** | 0.003 | 0.452 | 0.004 | 0.470 | **0.003** | **0.552** | 0.007 | 0.494 | 0.003 | **0.542** | 0.002 | 0.535 | 0.006 | 0.542 | 0.003 | **0.600** | **0.002** | 0.586 |
| | 336 | 0.006 | 0.424 | 0.009 | 0.358 | **0.003** | **0.507** | 0.005 | 0.445 | 0.005 | 0.488 | **0.004** | **0.567** | 0.005 | 0.490 | 0.003 | 0.485 | **0.002** | **0.525** | 0.005 | 0.526 | 0.005 | 0.480 | **0.002** | **0.578** |
| | 720 | 0.007 | 0.432 | 0.153 | 0.398 | **0.003** | **0.508** | 0.006 | 0.448 | 0.074 | 0.514 | **0.005** | **0.569** | 0.008 | 0.474 | 0.011 | 0.472 | **0.002** | **0.510** | 0.006 | 0.491 | 0.003 | 0.489 | **0.002** | **0.574** |

ground truth and prediction as our objective function for achieving phase shifting invariance. For the other frequencies, we use the norm of the prediction sequence to reduce the value of the Fourier coefficient. Consequently, this loss function keeps the temporal dynamics of the original time series (i.e., dominant frequencies) and enables noise robustness by reducing white noises in non-dominant frequencies. We achieve phase shifting invariance by optimizing the following loss function:

$$\mathcal{L}_{phase}(\mathbf{Y}, \hat{\mathbf{Y}}) = \begin{cases} ||\mathcal{F}(\mathbf{Y}) - \mathcal{F}(\hat{\mathbf{Y}})||_p, & \text{dominant freq.} \\ ||\mathcal{F}(\hat{\mathbf{Y}})||_p, & \text{otherwise} \end{cases} \tag{5}$$

where $|| \cdot ||_p$ is the $L_p$ norm. To obtain the dominant frequency terms, we calculate the norm of the Fourier coefficient for each frequency and filter them with the squared root of sequence length, $\sqrt{T'}$. We also guarantee the minimum number of dominant frequencies as $\sqrt{T'}$. This loss function obtains uniform amplification invariance through the application of a normalization technique to Fourier coefficients. For example, $\sin x$ and $c \cdot \sin x$ have the same Fourier coefficients if appropriately normalized. In summary, from Eq. 5, we can obtain (1) invariance for phase shifting, (2) invariance for uniform amplification, and (3) robustness to noise.

**Invariances with Autocorrelation (Uniform Amplification)**    Although Fourier coefficients can be considered a reasonable solution to determine the periodicity of the target time-series, they are not completely invariant to phase shifting for three reasons: (1) the data statistics (e.g., mean and variance) keep changing, (2) such changing statistics also cause changes in Fourier coefficients even at the same frequency, and (3) objectives only with a norm of Fourier coefficient cannot fully represent the original time-series. Thus, we introduce an objective based on normalized cross-correlation, which satisfies Eq. 2 for a periodic signal:

$$\mathcal{L}_{amp}(\mathbf{Y}, \hat{\mathbf{Y}}) = ||R(\mathbf{Y}, \mathbf{Y}) - R(\mathbf{Y}, \hat{\mathbf{Y}})||_p, \tag{6}$$

where $R(\cdot, \cdot)$ is a normalized cross-correlation function. This loss function helps predicted sequences mimic label sequences by calculating the difference between the autocorrelation of the label sequences and the cross-correlation between the label and predicted sequences. Therefore, the label and prediction have similar temporal dynamics, regardless of phase shifting or uniform amplification.

In summary, we introduce TILDE-Q, combining Eq. 4, Eq. 5, and Eq. 6 as follows:

$$\mathcal{L}_{TILDEq}(\mathbf{Y}, \hat{\mathbf{Y}}) = \alpha\mathcal{L}_{a.shift}(\mathbf{Y}, \hat{\mathbf{Y}}) + (1 - \alpha)\mathcal{L}_{phase}(\mathbf{Y}, \hat{\mathbf{Y}}) + \gamma\mathcal{L}_{amp}(\mathbf{Y}, \hat{\mathbf{Y}}), \tag{7}$$

where $\alpha \in [0, 1]$ and $\gamma$ are hyperparameters.

## 5 EXPERIMENTS

In this section, we present the results of our comprehensive experiments, demonstrating the effectiveness of TILDE-Q and the importance of transformation invariance.

Table 2: Experimental results of short-term time-series forecasting on the three datasets with sequence-to-sequence GRU model.

| Methods | GRU + MSE | | | | GRU + DILATE | | | | GRU + TILDE-Q | | | |
|---|---|---|---|---|---|---|---|---|---|---|---|---|
| Eval | MSE | DTW | TDI | LCSS | MSE | DTW | TDI | LCSS | MSE | DTW | TDI | LCSS |
| Synthetic | **0.0107** | 3.5080 | **1.0392** | 0.3523 | 0.0130 | 3.4005 | 1.1242 | **0.3825** | 0.0119 | **3.2873** | 1.1564 | 0.3811 |
| ECG5000 | 0.2152 | 1.9718 | 0.8442 | 0.7743 | 0.8270 | 3.9579 | 2.0281 | 0.4356 | **0.2141** | **1.9575** | **0.7714** | **0.7773** |
| Traffic | **0.0070** | 1.4628 | 0.2343 | 0.7209 | 0.0095 | 1.6929 | 0.2814 | 0.6806 | 0.0072 | **1.4600** | **0.2276** | **0.7220** |

**Experimental Setup**    We conduct the experiments with four state-of-the-art models–Informer (Zhou et al., 2021), N-Beats (Oreshkin et al., 2020), Autoformer (Wu et al., 2021), and FEDformer (Zhou et al., 2022)–and a simple sequence-to-sequence gated recurrent unit (GRU) model. For model training, we use seven real-world datasets–ECG5000, Traffic, ETTh2, ETTm2, ECL, Exchange, and Weather–and one synthetic dataset, Synthetic. We repeat each experiment with a model and dataset 10 times in combination with three different objective functions. Appendix A provides detailed explanations of the datasets, hyperparameter settings, model, and source code. We also provide experimental results with NSFormer (Liu et al., 2022) in Appendix.

**Evaluation Metrics**    In this experiment, we evaluate TILDE-Q with three evaluation metrics: mean squared error (MSE), dynamic time warping (DTW), and its corresponding temporal distortion index (TDI), all of which are referred from Le Guen & Thome (2019). As DTW is sensitive to noise and generates incorrect paths when one of the time-series data is noisy (as discussed in Sec. 3.3), we additionally use the longest common subsequence (LCSS) for comparison, which is more robust to outliers and noise (Esling & Agon, 2012). The longer the matched subsequences, the higher the LCSS score will be achieved in modeling the shapes. For state-of-the-art models, we report the MSE and LCSS. For detailed results, including those for DTW and TDI, please refer to Appendix B.

**Experimental Results and Analysis**    Table 2 shows the results of the short-term forecasting performance of the GRU model optimized with the MSE, DILATE, and TILDE-Q metrics. With the Synthetic dataset, each metric used shows its own benefits. This result indicates that loss functions with shape similarity or MSE have their specialty for shape and exact value, respectively. It also means a better MSE does not guarantee a better solution for temporal dynamics. Moreover, since the model is evaluated with real-world datasets, it is revealed that TILDE-Q outperforms other objective functions in most evaluation metrics. These results indicate that our approach to learning shapes in time-series data achieves better results than existing methods for forecasting. DILATE does not show impressive performance with ECG5000 due to its high sensitivity to noise, as discussed in Sec. 3.3. Table 1 summarizes the experimental results obtained with the four state-of-the-art models, N-Beats, Informer, Autoformer, and FEDformer. The models make predictions for both short-term ($L$=96) and long-term ($L$ up to 720). Thus, we can investigate their performances with different forecasting difficulties. In most datasets, the models with TILDE-Q outperform those with other training metrics. Especially for long-term forecasting, N-Beats and Informer with TILDE-Q show significantly improved performance compared to those with the other metrics. Appendix B presents some visual examples and more detailed analysis, qualitative experiments with example visualizations, and ablation study results. These results imply that TILDE-Q improves the performance of the models in learning temporal dynamics, including the LCSS of N-Beats (improved over 10%).

## 6    CONCLUSION AND FUTURE WORK

We propose TILDE-Q that allows shape-aware time-series forecasting in a timely manner. To design TILDE-Q, we review existing transformations in time-series data and discuss the conditions that ensure transformation invariance during optimization tasks. The designed TILDE-Q is invariant to amplitude shifting, phase shifting, and uniform amplification, ensuring a model better captures shapes in time-series data. To prove the effectiveness of TILDE-Q, we conduct comprehensive experiments with state-of-the-art models and real-world datasets. The results indicate that the model trained with TILDE-Q generates more timely, robust, accurate, and shape-aware forecasting in both short-term and long-term forecasting tasks. We conjecture that this work can facilitate future research on transformation invariances and shape-aware forecasting.

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
