# OpenReview forum: "TILDE-Q: A Transformation Invariant Loss Function for Time-Series Forecasting"
_ICLR.cc/2024/Conference — Submitted to ICLR 2024_

### Official Review · Reviewer_xHdk · 2023-10-24

**Soundness:** 2 fair
**Presentation:** 2 fair
**Contribution:** 2 fair
**Rating:** 5
**Confidence:** 4

**Summary:**

The authors proposed a loss function to account for distortions in the time series data. In particular, the loss function is designed to account for the amplitude and phase shifting in the prediction. The authors performed extensive numerical experiments to demonstrate advantages of the proposed method.

**Strengths:**

It looks relatively easy to apply the proposed method to improve existing time series models.  The authors performed extensive experiments using various combinations of the model and the proposed loss function.

**Weaknesses:**

The proposed loss function only deals with the amplitude and phase shifts in the prediction, unlike the various types of distortions described in the manuscript. The level of novelty and robustness of the manuscript seem below the standard of ICLR. Please, see the questions below.

**Questions:**

Major concerns:

1. The amplitude shifting loss seems not optimal for noisy time series data, which is almost all time series of interest. The MSE loss aims to estimate the expectation of the probability distribution of the time series, which is the optimal solution, while the amplitude shifting loss always introduces a bias so that the prediction is always above or below of the noisy signal.

2. Similar to question 1, in many cases, the amplitude shifting loss will compete against the standard MSE loss. For some problems, MSE loss will be optimal and, for the other cases, probably the amplitude shifting loss makes sense. However, how to decide which one to use?

3. The Fourier loss makes a much more sense than the amplitude shifting loss. It will capture the periodicity of the time series data. Recently, it has been shown in many studies that capturing a periodicity is a key element in time series forecasting. However, the authors should elaborate how they compute the Fourier coefficients. For example, when if the signal is not periodic, the Fourier series do not converge, meaning it is difficult to truncate the Fourier series. So, deciding the dominant frequency is non-trivial. What's the theoretical reason to choose $\sqrt{T'}$ to truncate the Fourier seires?

4. The loss based on the auto-correlation also makes a sense. However, I don't fully understand the explanation of the weakness of the Fourier loss. The authors explained three reasons why the Fourier method fails to represent the characteristics of the time series. But all of the three reasons exactly apply the same for the auto-correlation function. As the authors explained, if the time series is non-stationary, the auto-correlation also keeps changing, like the Fourier series. As a matter of fact, auto-correlation and Fourier series are like two sides of a coin.

5. What's the logic behind of the structure of the loss function in (7)? Why is $L_{phase}$ grouped with $L_{shift}$? And $L_{amp}$ is separate?

6. It may be interesting to compare the results with DLinear (https://arxiv.org/pdf/2205.13504.pdf), where they proposed a very simple linear model with trend and seasonality decomposition, which should be similar to the loss functions proposed.

---

> ### Author Response · Authors · 2023-11-22
> **Official Response to the Reviewer xHdk**
>
> Q1. The amplitude shifting loss seems not optimal.
> > Thank you for your valuable feedback. For clarification, we would like to point out that having a non-optimal solution is often better than an optimal solution as prior work shows [1, 2]. Ishida et al. [1] revealed that providing some “tolerance” for the loss function makes training easier, resulting in better generalization. We also want to emphasize that the noise-robustness of our softmax loss depends on the choice of the distance function. We described and emphasized these issues in Appendix A.
>
> Q2. It seems like $L_{a.shift}$ competes against MSE. How do we decide which one to use?
> > We think that the amplitude shifting is not designed to perfectly replace the standard MSE. Rather, amplitude shifting can be used as a complementary technique (e.g.,  [1, 2]) that allows tolerance for the models that do not necessarily aim at zero distance.  As a result, it achieves greater generalization than MSE, therefore, $L_{a.shift}$ could be used for all the datasets. In Appendix C, we provide multiple qualitative examples to illuminate this issue.
>
> Q3. Questions about design rationale of $L_{phase}$
> > In our design of $L_{phase}$, we chose the dominant frequency with $\sqrt{T’}$, which is equivalent to the statistical significance (i.e., one standard deviation) on Fourier coefficients. Furthermore, $L_{phase}$ guarantees the minimum number of dominant frequencies as $\sqrt{T’}$ when handling non-periodic signals. This regularization works as a noise filter, which is helpful in practice (detailed information is available in Appendix A). For an in-depth validation of the noise-robustness of our method, we provide qualitative examples in Appendix C.2.
>
> Q4. $L_{phase}$ and $L_{amp}$ seem to be monotonic. Do they actually have differences?
> > If we used raw Fourier coefficients and correlation computation, your comment would be correct. However, we did not use the raw coefficients and correlation as they are. For clarification, 1) we regularized non-dominant terms in Fourier coefficients and 2) we utilize a "normalized version" of full correlation, which is invariant to the uniform amplification (since it is normalized by the norm of signal). We have clarified these issues in Appendix A.
>
> Q5. Design rationale for $\alpha$ and $\gamma$
> > We introduce $\alpha$ and $1 - \alpha$ for $L_{a.shift}$ and $L_{phase}$ since they complement each other in terms of periodicity and statistical differences (in mean value and variance). $L_{amp}$ acts as a supplement, so we set $\gamma = 0.01$.  Note that we have provided detailed information on the setting and our source code in Appendix B. We have also described the design rationale for each sub-loss, $\alpha$, and $\gamma$ in Appendix A.
>
> Q6. It may be interesting to compare the results with DLinear
> > Thank you for your comment. You can check the experimental results of NLinear in Anonymized GitHub (link available in Appendix B). In short, in most cases, MSE and TILDE-Q show comparable performance. However, we find that in the case of the Weather and Exchange datasets, which are hard to forecast due to frequent fluctuation patterns, TILDE-Q achieves better performance. In the case of the Weather dataset, we observe 6% to 12% improvements for NLinear.
>
> [1] Do we need zero training loss after achieving zero training error? ICML 2020
>
> [2] WaveBound: Dynamic Error Bounds for Stable Time Series Forecasting, NeurIPS 2022

---

### Official Review · Reviewer_Ba52 · 2023-10-28

**Soundness:** 3 good
**Presentation:** 2 fair
**Contribution:** 2 fair
**Rating:** 6
**Confidence:** 4

**Summary:**

This work introduces a new loss function for time series forecasting models, TILDE-Q. TILDE-Q is a combination of three terms designed to penalize predictions while preserving invariance to one specific "distortion" (each): amplitude shifting (constant value shifting), phase shifting (constant translation), and amplitude scaling.

**Strengths:**

The method is well-motivated and intuitive, and the evaluation is thorough, including most common deep learning time series baseline methods and tasks.

**Weaknesses:**

My main reservations with this work are limited novelty (softmax and frequency domain losses are arguably the most basic type of objective in forecasting), and the work does not spend any time on the analysis of the proposed method, or on providing further insights. As an example, comments about softmax-ratios being invariant to constant shifts could be expanded with additional results, or providing a discussion on other ways to achieve invariance to constant scaling. The same goes for the caveats related to computing a loss in frequency domain (Sec 4.2, paragraph 2) and choice of dominant Fourier coefficients, which is not at all trivial in practice, especially for varied datasets.

**Questions:**

* I don't see how the softmax loss can satisfy your eq (1) requirement; the optimal solution where $d(y_i, \hat y_i) = k$ will result in a non-zero $\mathcal{L}_{a,shift}$. Could the authors clarify on how the softmax helps strengthen value shift invariance?
*  How do you choose the dominant frequency-domain coefficients for $\mathcal{L}_{phase}$? Does regularizing "non-dominant" frequencies for noise robustness help in practice? How do you deal with the fact that time series are finite-length signals (and there could be boundary effects)?
* How do you choose $\alpha$ and $\gamma$? Why stop at three terms in the loss function? Did you empirically find these distortions to be most common in the datasets of interest?
* It would have been valuable to carry out an analysis and see how common these distortions are in the datasets you evaluate on. It is difficult to assess significance otherwise (other than looking at the aggregate metrics)

---

> ### Author Response · Authors · 2023-11-22
> **Official Response to the Reviewer Ba52**
>
> Thank you for your valuable comments! We revised our manuscripts and appendices accordingly. Below are the detailed responses.
>
> Q1. Does $L_{a.shift}$ really satisfy a requirement (i..e, Eq. (1))?
> > Yes, $L_{a.shift}$ satisfies a requirement. In short, we reformulated the “ensuring equal gaps between all points” into “making a uniform distribution of the gaps (i.e., $\sum_i{p_{d_i}\log p_{d_i}}$),” which is equivalent to entropy maximization. Therefore, we used the Softmax function for $L_{a.shift}$ and $L_{a.shift}$ satisfies Eq. 1. A detailed explanation of the sub-loss design can be found in Appendix A.
>
> Q2. Could you explain more about your design rationale for $L_{phase}$, especially for the dominant frequency term selection?
> > In our design, we chose the dominant frequency with $\sqrt{T’}$, which is equivalent to the statistical significance (i.e., one standard deviation) on Fourier coefficients. Furthermore, $L_{phase}$ guarantees the minimum number of dominant frequencies as $\sqrt{T’}$ when handling non-periodic signals. This regularization works as a noise filter, which is helpful in practice (detailed information is available in Appendix A). For the validation of our noise-robustness, we also provide qualitative examples in Appendix C.2.
>
> Q3. How do you choose $\alpha$ and $\gamma$? Could you clarify the design rationale?
> > (Why three terms? Is it empirically selected?) In Sec. 4.1., we describe why we chose those three distortions. We excluded uniform time scaling, dynamic time scaling, and dynamic amplification, as they either require data-specific hyperparameters (e.g., a window size of DTW) or can impede the proper encoding of information for forecasting (e.g., causing a failure to encode periodicity).
>
> > (How do you choose alpha and gamma?) We introduce $\alpha$ and $1 - \alpha$ for $L_{a.shift}$ and $L_{phase}$ since they complement each other in terms of assumption on periodicity and the statistical differences (in mean value and variance). $L_{amp}$ acts as a supplement, so we set $\gamma = 0.01$.  Note that we have provided detailed information on the setting and our source code in Appendix B. We have also described the design rationale for each sub-loss, alpha, and gamma in Appendix A.
>
> Q4. Do distortions always exist among all datasets? How common these distortions are?
> > As we described in Sec. 3.1, distortions are defined between two time-series, which means they are the gap between two similar time-series. We want to emphasize that distortion itself can appear between ground-truth time-series and prediction results regardless of the dataset. If we ignore the potential impact of the distortion and keep using an objective function, which is vulnerable to distortion, the model produces uninformative forecasting results, as shown in the top and middle parts of Fig. 1. For a more thorough explanation and evaluation of the model and cases, we provide qualitative results in Appendix C.

---

### Official Review · Reviewer_gjfT · 2023-10-31

**Soundness:** 2 fair
**Presentation:** 2 fair
**Contribution:** 2 fair
**Rating:** 5
**Confidence:** 4

**Summary:**

This work investigates distortion handling in time series forecasting problems, specifically introducing a novel loss function to guide models in considering amplitude, phase, and uniform amplification shifting invariance.

**Strengths:**

1. The related work is comprehensive and shows a thorough analysis of different time series distortions.

**Weaknesses:**

1. It appears that the authors did not provide the complete version of the paper, as the appendices are missing.
2. The experiments lack recent time series forecasting models.
3. The experiments are not complete.
4. The experiments lack detailed analysis.

**Questions:**

The paper is not complete since some important appendices related to experiments are missing.
1. I'm interested in seeing additional results using TILDE-Q on recent time series forecasting models, including NLinear, DLinear, Scaleformer, PatchTST, Depts, N-hits, and SCINet.
2. More in-depth analysis of the primary results are expected.

---

> ### Author Response · Authors · 2023-11-15
> **About appendix**
>
> Thank you for your comment.
>
> However, we kindly remind you that we have already uploaded the appendix in our submission. Please check supplementary materials.
> In the appendix, we provide 1) recent time-series forecasting models, including NSFormer (AAAI'22). We also plan to update the results with NLinear and DLinear. Furthermore, the appendix provides the detailed experimental setting, results, and ablation study.
>
> We are looking forward to you checking the appendix and updating the review. We will make our best efforts to answer your questions and update the manuscripts with your feedback.

---

> ### Author Response · Authors · 2023-11-22
> **Official Response to the Reviewer gjfT**
>
> Thank you for your comment. We feel sorry that you missed the appendices that we have provided in the original submission. To help you find the information that you are looking for, we explain the location of the information.
>
>  - The experimental results with recent time-series forecasting models, including NSFormer (AAAI’22) and NLinear (AAAI’23) are available in Appendix C and Anonymized Github (through a link provided in Appendix B), respectively.
>
> - We describe the detailed experimental setting in Appendix B. The experimental results and their analysis are in Appendices C.1– C.3, while the qualitative evaluation is in C.2.

---

### Official Review · Reviewer_sFtV · 2023-11-02

**Soundness:** 3 good
**Presentation:** 3 good
**Contribution:** 3 good
**Rating:** 8
**Confidence:** 4

**Summary:**

A shape aware loss function, TILDE-Q, is introduced to capture distortios between time-series signals. TILDE-Q aims to be invariant to shift and scale distortions, both in space and time, to capture, e.g., shifts in phase and amplitude. Compared to traditional losses, such as MSE and DILATE, the proposed TILDE-Q loss helps various models to generate more accurate predictions on numerous datasets.

**Strengths:**

_Originality:_ The conducted analysis reveals relevant aspekts for time-series forecasting and the proposed method seems to reasonably compensate the limitations of conventional loss functions. You may want to add temporal convolution networks (TCN) from [Lea et al. (2016)](https://link.springer.com/chapter/10.1007/978-3-319-49409-8_7) and [Kalchbrenner et al. (2016)](https://arxiv.org/abs/1610.10099) to your related work section. Otherwise, the manuscript adequately cites related work.

_Quality:_ The writing, presentation, and demonstrations are well approachable and claims are properly supported by exhaustive experimental results. Some comments about limitations of TILDE-Q would be highly appreciated.

_Clarity:_ The organization of the manuscript is appealing and the arguments are well presented and iteratively constructed. In general, the manuscript provides the necessary information to understand the core message and to follow the clear line of argumentation.

_Significance:_ Results demonstrate the effectiveness of the proposed method and have the potential to provide valuable insights to the time-series forecasting community. Given the computational overhead is not too large, the proposed loss could evolve to an alternative to traditional loss functions, especially replacing MSE.

**Weaknesses:**

1. Improvements often seem to be marginal only (even though the score in a metric does not necessarily correlate with the actual quality of the qualitative forecast). Since you report that 10 models were trained, it would be easier to assess the loss quality if you provide error variations in form of $\sigma$ scores.
2. Given the various computations required to calculate the TILDE-Q loss, it is unclear how much computational overhead the method introduces and whether the benefits outweigh and justify the increased time complexity.

**Questions:**

1. Could TILDE-Q serve as a performance measure itself too? The challenge of quantifying the skill of a time-series forecast is ubiquous. If the TILDE-Q score matches closely with visual inspections, this would be agreat metric as well. It would be a great outcome, if the judgement of subjects on the similarity of two time-series would correlate strongly with TILDE-Q.
2. Inspecting the qualitative results in the appendix, can you formulate situations in which TILDE-Q performs poorly (consistently)?

---

> ### Author Response · Authors · 2023-11-22
> **Official Response to the Reviewer sFtV**
>
> Thank you for your feedback! We clarified the issues in our manuscripts and appendix. We would be grateful if our revision resolves your questions. Below are detailed responses.
>
> W1. About $\sigma$ score and in-depth evaluation of our results
> > We uploaded the results with the $\sigma$ score in our Anonymized GitHub, which is also provided in Appendix B. Please note that we only provide $\sigma$ scores in GitHub due to their large size. Additionally, we have updated various qualitative results and their analysis in Appendix C.
>
> W2. Lack of justification for TILDE-Q's computational costs and its benefits
> > We updated our design rationale and discussion of time complexity in Appendix A. In short, TILDE-Q only requires O(n log n) time for its computation, which has an almost negligible impact on the overall time complexity.
>
>
> Q1. Could TILDE-Q serve as a performance measure?
> > Yes. We think that if TILDE-Q is correctly used, it could work as a performance measure, as we show in the results, which will create a major impact on the domain. In addition, researchers could formulate their own measurements if their measurements met the requirements we described in Sec. 4.1.
>
> Q2. Can you formulate situations in which TILDE-Q performs poorly (consistently)>
> > Thank you for your comment. We updated Appendix C, adding descriptions of the current stage of TILDE-Q with qualitative examples.

---

### Author Response · Authors · 2023-11-22
**Response to Common Questions**

1. Design rationale for the TILDE-Q hyperparameters

> We introduce $\alpha$ and $1 - \alpha$ for $L_{a.shift}$ and $L_{phase}$ since they complement each other in terms of periodicity and statistical differences (in mean value and variance). $L_{amp}$ acts as a supplement, so we set $\gamma = 0.01$.
For detailed information on the hyperparameter setting, we provide 1) the detailed setting and 2) the source code in Appendix B, and we provide 3)  detailed design rationales for each sub-loss, $\alpha$, and $\gamma$ in Appendix A.


2. Detailed explanation for sub-losses, especially for the amplitude shifting and phase shifting losses

> When designing $L_{a.shift}$, we reformulated the “ensuring equal gaps between all points” into “making a uniform distribution of the gaps (i.e., $\sum_i{p_{d_i}\log p_{d_i}}$),” which is equivalent to entropy maximization. Therefore, we used the Softmax function for $L_{a.shift}$. A detailed explanation of the sub-loss design can be found in Appendix A.

> In designing $L_{phase}$, we chose the dominant frequency using $\sqrt{T’}$, which represents the statistical significance of Fourier coefficients. Furthermore, $L_{phase}$ guarantees the minimum number of dominant frequencies as $\sqrt{T’}$ when applied to non-periodic signals. This regularization also works as a noise filter, which further improves performance in practice, as described in Appendix A. For an in-depth validation of the noise-robustness of our method, we also provide qualitative examples in Appendix C.2.

3. Comparisons to SOTA models–NSFormer and NLinear

> Thank you for your comment. We have updated our comparison results with two more SOTA models. First, we uploaded the experimental results of NLinear in Anonymized GitHub (a link is available in Appendix B). We also report the experiment results with NSFormer (AAAI’22) in Appendix C.

> The new experimental results suggest that MSE and TILDE-Q show comparable performance in most cases. However, when there are many fluctuations in the datasets (i.e., difficult parts), TILDE-Q outperforms other models, as shown with the Weather and Exchange datasets. In particular, TILDE-Q showed 6% to 12% improvement compared to NLinear for the Weather dataset.

---

### Meta-Review · Area_Chair_kgsw · 2023-12-08

**Metareview:**

The paper introduces and motivates a new shape-aware loss function to better capture the importance of shape in time series forecasting compared to standard point forecasting losses like MSE. The shape aware loss function is motivated by create new loss terms that are invariant to various types of distortions. The paper conducts experiments on real-world datasets and showcase gains in both MSE and shape-aware metrics by simply switching out the loss function.

This was a borderline paper. It has interesting contributions, and brings out valuable insights around the need to predicting the right shape of the forecasts that current point losses cannot capture. The loss function construction is well presented and motivated, and the empirical results seem sound. However some reviewers had a concern around whether the empirical evaluation were thorough enough. The models used in this paper are missing several of the recent, state-of-the-art models (PatchTST, TimesNet etc) and from the new results it seems like the gains obtained by this loss function varies quite a bit with the choice of  model - for example, the gains due to the loss-function on NLinear were quite small.  Additionally, multiple reviewers commented on limited novelty given that the shape-aware loss mostly deals with amplitude and phase shift distortions (while the authors address why some of the other distortions werent captured in the loss - a more detailed empirical analysis would have been useful)
Nevertheless the paper address an important problem and I would urge the authors to strengthen the empirical evaluation and resubmit

**Justification For Why Not Higher Score:**

The empirical evaluation falls marginally short of being strong or rigorous enough to meet the ICLR bar (see text in the "Addditional Comments" section above)
-The models used in evaluation were not state-of-the-art, and upon reporting (in the rebuttal) results using a more recent model, the gains in the paper seem to have diminished significantly.
-The evaluation was conducted on a limited set of public datasets, which opens some questions around cherry-picking of the datasets. For he paper used six of the  standard Autoformer/Informer LTSF datasets, but left out three others that are also part of the same collection.

**Justification For Why Not Lower Score:**

N/A

---

### Decision · Program_Chairs · 2024-01-16

Reject